

# A methodological and reporting quality assessment of systematic reviews/meta-analyses on exercise interventions for cognitive function in older adults with mild cognitive impairment

Wanli Zang[1], Qinghai Zou[2], Ningkun Xiao[3], Mingqing Fang[4], Su Wang[1] and Jingjing Chen[5]

[1] Harbin Sport University, Harbin, China
[2] Physical Education Department, Harbin Engineering University, Harbin, China
[3] Department of Psychology, Ural Federal University, Yekaterinburg, Russia
[4] Xiangya Hospital, Central South University, Changsha, China
[5] School of Basic Medicine, Changchun University of Chinese Medicine, Jilin, China

Corresponding authors
Su Wang, wangsu@hrbipe.edu.cn
Jingjing Chen,
15568772687@163.com

## ABSTRACT

**Objective**. To assess the methodological quality of meta-analytic literature on exercise interventions for cognitive function in patients with mild cognitive impairment (MCI) and the certainty of evidence for its outcome indicators, and to provide clinicians and researchers with more reliable data for making decisions.

**Methods**. Meta-analytic literature related to the effect of exercise intervention on cognitive function in patients with mild cognitive impairment was searched through PubMed, Cochrane Library, Embase, Scopus, Physiotherapy Evidence Database and Web of Science, all with a search period frame of each database until June 1, 2024. The AMSTAR2 scale was used to evaluate the methodological quality of the included studies.

**Results**. Seventeen meta-analyses were included. The AMSTAR2 scale evaluation results showed that there was one medium-quality studies (5.55%), seven low-quality studies (38.88%), and 10 very low-quality studies (55.55%). Methodological deficiencies included failure to prepare a plan and provide a registration number, literature screening, data extraction, reasons for exclusion not described in detail, poor implementation process for systematic evaluation, and failure to describe the source of funding for the included studies or relevant conflicts of interest.

**Conclusion**. The overall methodological quality of the meta-analytic literature is low, and the certainty of evidence is low. We encourage the conduction of high-quality randomized trials to generate stronger evidence. Subsequent systematic reviews can then synthesize this evidence to inform future research and clinical guidelines.

## INTRODUCTION

Mild cognitive impairment (MCI) is characterized by early progressive cognitive decline that lies between normal cognitive decline due to aging and early degeneration of dementia (*Knopman & Petersen, 2014*; *Sachdev et al., 2014*). MCI is considered a prodromal stage of Alzheimer's disease (AD), but does not meet the diagnostic criteria for AD (*Jicha et al., 2010*; *Jongsiriyanyong & Limpawattana, 2018*). Abnormal brain functional activity in MCI is associated with abnormal deposition of amyloid-beta protein, which causes homeostatic imbalance in brain functional network activity by impairing inhibitory gamma-aminobutyric acid (GABA) neurons or increasing synchronization of excitatory synaptic activity (*Mormino et al., 2011*; *Zott & Konnerth, 2023*). Additionally, an imbalance in the phosphorylation and dephosphorylation of tau proteins in the brain can lead to hyperphosphorylation, forming double-helix filaments and neuronal fiber aggregation, impairing synaptic and neuronal functions, and ultimately affecting cognitive function (*Krance et al., 2019*).

According to statistics, approximately 15.56% of community members aged 50 years and older worldwide experience MCI (*Bai et al., 2022*). As the global population ages and life expectancy increases, the incidence of MCI is gradually rising (*Koyanagi et al., 2018*). MCI is a high-risk group for Alzheimer's disease, which is currently the fourth leading cause of death in the older adults, after heart disease, tumors, and stroke. It is one of the most significant global public health and social care challenges facing humanity both today and in the future. No single medication can halt or reverse the MCI disease process (*Eshkoor et al., 2015*). Delayed treatment in the preliminary stages of the illness may lead to worsening of cognitive function, causing a significant burden on individuals, their families, and society (*Petersen et al., 2018*). Consequently, complementary therapies and non-pharmacological approaches are also gaining attention among researchers, who are gradually shifting their MCI management strategies toward non-pharmacological therapies (*Gauthier et al., 2010*; *Simon, Yokomizo & Bottino, 2012*).

Exercise is increasingly recognized as a non-pharmacological intervention for MCI (*Teixeira et al., 2012*; *Wang et al., 2020*). For example, exercise can activate the PI3K/Akt/mTOR signaling pathway to induce autophagy in brain cells, allowing them to reduce the level of amyloid-beta aggregates in the brain through the autophagy system (*Heras-Sandoval et al., 2014*; *Ma et al., 2013*). Exercise may also protect the hippocampus by preventing neuronal death and limiting the production of inflammatory markers (*Małkiewicz et al., 2019*). Exercise promotes the formation and survival of new neurons in the hippocampal region, thereby enhancing plasticity and memory function in the hippocampus (*Curlik 2nd & Shors, 2013*). In recent years, there has been a growing body of clinical research investigating the effects of exercise therapies on cognitive performance in older adults with MCI as well as numerous meta-analyses in this area. Such studies enable the systematic and comprehensive presentation of all available evidence while improving the accuracy of the data by reducing bias from random errors and decreasing random errors generated by individual studies.

Several meta-analyses demonstrated the efficacy of exercise in improving cognitive function in older adults with MCI. However, the quality of these meta-analyses varies widely, and few studies have thoroughly assessed the level of evidence, leaving its certainty largely unclarified. In a related effort, *Venegas-Sanabria et al. (2021)* conducted an umbrella review to assess the impact of physical activity on cognitive domains in patients with dementia and mild cognitive impairment. Utilizing the AMSTAR2 tool, they evaluated the therapeutic effectiveness of exercise interventions in these populations. Notably, their review targeted two distinct groups and included only 11 studies, a relatively small sample that underscores the challenges in drawing broad conclusions.The primary focus of their research was to explore the efficacy of exercise interventions, rather than scrutinizing the methodological robustness of previous meta-analyses in this area. Evaluating the methodological and certainty of evidence can guide users by synthesizing the findings of meta-analyses at a higher level. This study aimed to reassess the methodological quality and applicability of systematic reviews of exercise therapy in older adults with MCI by assess the meta-analysis literature on the impact of exercise interventions on cognitive function in older adults with MCI using the AMSTAR2 (*Leclercq et al., 2020*). This assessment seeks to understand the status and existing issues of evidence-based research on the cognitive function of older adults with exercise intervention in MCI and provides a reference for conducting high-quality clinical research on exercise therapy for older adults with MCI to guide clinical decision-making.

## METHODS

The protocol for this study was officially registered with INPLASY under the registration number 2023100065. The associated DOI for this registration is 10.37766/inplasy2023.10.0065.

### Search strategy

Two independent investigators (WZ, NX) performed comprehensive literature searches in five pertinent databases: PubMed, Cochrane Library, Embase, Scopus, Physiotherapy Evidence Database and Web of Science. The search time limit was set from the inception of each database to June 1, 2024. The search terms used included: ("Exercise"[Mesh] OR "exercise" OR "physical activity" OR "resistance training" OR "strength" OR "endurance" OR "walking" OR "yoga" OR "TaiChi") AND ("Cognitive Dysfunction"[Mesh] OR "mild cognitive impairment" OR "MCI") AND ("systematic review" OR "meta-analysis") NOT ("Alzheimer's disease" OR "AD").

### Inclusion and exclusion criteria
#### Inclusion criteria

We incorporated relevant meta-analyses based on randomized controlled trials (RCTs) to examine the impact of exercise interventions on cognitive function in individuals with MCI. All participants included in the studies were required to have a clinically confirmed MCI diagnosis or met at least one of the following criteria: (1) self-reported memory loss or cognitive decline that disrupts daily functioning, which is not supported

by corroborating evidence from family members; (2) age-inappropriate memory deficits, as evidenced by performance on memory and cognitive scales falling below 1.5 standard (*Aarsland et al., 2009*) deviations from age- and education-matched norms (for example, Wechsler Memory Scale (WMS): 60–79 points; Mini-Mental State Examination (MMSE): 24–27 points; Montreal Cognitive Assessment (MoCA): 14–25 points); (3) intact general cognitive function except for memory, with preserved ability to perform activities of daily living; (4) absence of a dementia diagnosis, and insufficient evidence based on physician reports, patient history, and mental status examinations to diagnose Alzheimer's disease (AD) or other types of dementia. The included meta-analyses were required to assess interventions involving at least one exercise group, with no restrictions on the specific type of exercise employed, focusing specifically on global cognition as the primary outcome. The control groups were limited to those who did not participate in exercise, engaged only in low-intensity activities such as stretching, or participated in conventional treatment, health education, or a no-intervention control condition. This design allowed for a comprehensive assessment of the effects of various exercise modalities on cognitive function in comparison to non-exercise or minimal-intervention control groups.

### Exclusion criteria

Studies were excluded if they met any of the following criteria: (1) non-English language publications; (2) conference proceedings, abstracts, or presentations; (3) duplicate publications; (4) Cochrane meta-analyses in the planning or title registration phase; (5) participant populations consisting of AD patients; (6) intervention groups involving combined exercise and cognitive training, music therapy, or other non-exercise-based modalities; (7) no availability of full-text or extractable data; (8) animal studies; and (9) outcome measures not directly related to cognitive function.

## Literature selection and data extraction

Two investigators (WZ, NX) independently screened and organized the literature and extracted relevant information. The selection process initially involved reviewing the titles and abstracts, followed by cross-checking the extracted data. In cases of disagreement, a third investigator (MF) assisted in the decision-making process. After excluding irrelevant literature through preliminary screening, the full text of the remaining articles was reviewed for final inclusion. The following data were extracted: (1) basic information, such as study title, authors, publication date, journal, number of original studies included, sample size, intervention details, outcome measures, main conclusions, quality assessment tools, meta-analysis registration numbers, and funding sources; (2) methodology for literature evaluation and reporting quality-related content; and (3) qualitative or quantitative results for each outcome measure. Primary outcome measures included the Mini-Mental State Examination (MMSE), Alzheimer's Disease Rating Scale-Cognitive Subscale (ADAS-cog), and Montreal Cognitive Assessment (MoCA).

## Methodological quality evaluation

Two researchers (WZ, NX) independently assessed the methodological quality of the included studies using the AMSTAR2 scale, which comprises 16 items. Each item was

rated as "yes," "no," or "partially yes" based on the extent to which evaluation criteria were satisfied. Items 2, 4, 7, 9, 11, 13, and 15 are considered critical. Studies were assigned one of four credibility ratings: "High," "Moderate," "Low," or "Very Low." A rating of "High" was given if a study had 0 or 1 non-critical flaws, "Moderate" if it had more than one non-critical flaw, "Low" if it had one critical flaw with or without non-critical flaws, and "Very Low" if it had more than one critical flaw. Disagreements during the evaluation process were resolved through discussions with a third(MF) researcher.

# RESULTS

## Literature screening process and results

Eighteen meta-analyses were ultimately included in the analysis after screening 1,762 articles based on specified criteria. Figure 1 illustrates the literature screening process. The list of references deleted after reading the full text can be found in the Appendix S1.

## Basic characteristics of the included literature

The included meta-analyses were published between 2018 and 2023, and all the original studies were randomized and controlled. The largest study included 2,526 participants and the smallest had 182 participants, involving a total of 18,461 individuals. The interventions in the trial groups consisted of aerobic, physical, mental, resistance, and multipart exercises. The control group in the randomized trials maintained their regular daily activities without any specific exercise interventions. Regarding methodological quality assessment tools, 12 studies utilized the Cochrane Collaboration Network RoB risk assessment tool, four employed the PEDro scale, one used the Cochrane Collaboration Network RoB and PEDro, and one applied the EPHPP for risk assessment. All 18 studies consistently indicated that exercise effectively improves cognition in older adults with MCI. The basic characteristics of the included studies are presented in Table 1.

## Methodological quality evaluation of the included literatures

The methodological quality of the 18 included meta-analyses was assessed using the AMSTAR2 scale. The results showed that, in terms of quality, one study was rated as 'moderate', seven were rated as 'low', and ten were rated as 'very low'. The detailed results are presented in Table 2.

## Evaluating the impact of exercise interventions on cognitive assessment scores in mild cognitive impairment

### MMSE

The MMSE is a concise tool for assessing cognitive impairment, offering ease of administration, short duration, and objective results. It is widely used in the clinical, research, and community settings. However, it has limitations such as susceptibility to factors including education level, age, and cultural background, and low sensitivity to mild or subclinical cognitive impairment. Fifteen studies (Biazus-Sehn et al., 2020; Ahn & Kim, 2023; Law et al., 2020; Song et al., 2022; Gómez-Soria et al., 2022; Han et al., 2023; Lin, Chen & Cheng, 2023; Yuan, Li & Liu, 2022; Zhang et al., 2019; Zhou et al., 2020; Zhou & Li, 2022; Zhu et al., 2020; Zou et al., 2019; Pisani et al., 2021; Cai et al., 2023) examined the effects of

**Table 1  Basic characteristics of the literature included in the study.**

| Inclusion in the literature | Number of included studies/ sample size | Type of included study | Intervention | | Outcomes | Methodological quality assessment tools | Conclusion |
|---|---|---|---|---|---|---|---|
| | | | **Experimental group** | **Control group** | | | |
| *Ahn & Kim (2023)* | 21/1,916 | RCT | Aerobic, resistance, multicomponent, and neuromotor exercises | No treatment, usual treatment, usual care, health education and stretching | ① | Cochrane Collaboration RoB | The effect of exercise therapy on reducing cognitive function in elderly patients with mild cognitive impairment. |
| *Biazus-Sehn et al. (2020)* | 18/1,473 | RCT | Sport | No treatment | ①②③④⑤ | Cochrane Collaboration RoB | Physical exercise improves cognitive deficits in older patients with mild cognitive impairment. |
| *Gómez-Soria et al. (2022)* | 8/592 | RCT | MNPI | Passive (no intervention) or active control | ② | PEDro | Multicomponent non-pharmacological intervention (MNPI) improves overall cognitive performance in elderly MCI patients. |
| *Law et al. (2020)* | 35/2,079 | RCT | Physical exercise | No intervention/-placebo | ② | PEDro | Physical activity may reduce overall cognitive decline and behavioral problems in people with mild cognitive impairment or dementia. |
| *Pisani et al. (2021)* | 17/1,224 | RCT | Physical exercise | No | ①② | PEDro | Physical activity improved the cognitive abilities of MCI patients. |
| *Song et al. (2018)* | 7/620 | RCT | Physical exercise | All types of controls | ①②③ | EPHPP | Physical exercise, especially aerobic exercise, is beneficial to the overall cognitive ability of MCI patients. |

Zang et al. (2024), *PeerJ*, DOI 10.7717/peerj.17773

**Table 1** (*continued*)

| Inclusion in the literature | Number of included studies/ sample size | Type of included study | Intervention | | Outcomes | Methodological quality assessment tools | Conclusion |
|---|---|---|---|---|---|---|---|
| | | | **Experimental group** | **Control group** | | | |
| *Song et al. (2022)* | 7/854 | RCT | Traditional Chinese Medicine (Tai Chi, Baduan Brocade and Qigong) | All types of controls | ②③ | Cochrane Collaboration RoB | Tai Chi Promising as Alternative Mind-Body Intervention for MCI Rehabilitation in Elderly Patients. |
| *Yu et al. (2021)* | 10/709 | RCT | Baduan Brocade/Baduan Brocade + conventional treatment | Conventional treatment | ③ | Cochrane Collaboration RoB | Compared to conventional treatment, Baduanjin plus conventional treatment significantly improved cognitive and memory function in patients with mild cognitive impairment. |
| *Yuan, Li & Liu (2022)* | 12/820 | RCT | Dance activity, dance practice or dance therapy | Unlimited/any type | ②③ | Cochrane Collaboration RoB | Dance activities significantly improve overall cognition, memory, visuospatial function, cognitive flexibility, attention and balance in older patients with mild cognitive impairment. |
| *Zhang et al. (2019)* | 4/737 | RCT | Chinese traditional sports | Do not receive any intervention, routine care or other sports that are different from traditional Chinese sports | ② | Cochrane Collaboration RoB | Traditional Chinese exercise is associated with significant improvements in visuospatial function, but does not affect specific cognitive domains. |

Zang et al. (2024), *PeerJ*, DOI 10.7717/peerj.17773

**Table 1** (*continued*)

| Inclusion in the literature | Number of included studies/ sample size | Type of included study | Intervention | | Outcomes | Methodological quality assessment tools | Conclusion |
|---|---|---|---|---|---|---|---|
| | | | **Experimental group** | **Control group** | | | |
| *Zhang et al. (2020)* | 5/182 | RCT | Any form and intensity of external resistance training | Lifestyle routines without any movement, balance and tone exercises, and sham training similar to resistance training | ①②③ | Cochrane Collaboration RoB | Resistance training can improve general cognitive function and can be used to slow cognitive decline in MCI patients. |
| *Zhou et al. (2020)* | 8/428 | RCT | Exercise intervention (aerobic, resistance or multi-component exercise) | Sham exercise (such as stretching), placebo or no treatment or health education | ①②③ | Cochrane Collaboration RoB, PEDro | Exercise intervention significantly improved overall cognitive function. |
| *Zhou & Li (2022)* | 7/430 | RCT | Exercise interventions are not limited to exercise type, intensity, or duration | Other control interventions | ② | Cochrane Collaboration RoB | Exercise training has a positive effect on cognitive function in patients with mild cognitive impairment. |
| *Zhu et al. (2020)* | 3/524 | RCT | Aerobic dance | Health education and/or exercise, not aerobic dance training | ①② | Cochrane Collaboration RoB | Aerobic dance significantly improves overall cognitive function in older adults with mild cognitive impairment. |

**Table 1** (*continued*)

| Inclusion in the literature | Number of included studies/ sample size | Type of included study | Intervention | | Outcomes | Methodological quality assessment tools | Conclusion |
|---|---|---|---|---|---|---|---|
| | | | **Experimental group** | **Control group** | | | |
| *Zou et al. (2019)* | 9/1,105 | RCT | Mind-body movement | Positive (such as educational programs, memory training, physical exercise) or passive comparisons (such as waiting lists or unchanged lifestyles) | ②③ | PEDro | Physical and mental exercise has the potential to improve various cognitive functions in patients with mild cognitive impairment. |
| *Han et al. (2023)* | 18/1,700 | RCT | Exercise interventions are not limited to exercise type, intensity, or duration | Controls could be exercises of stretching, activities of health education, routine care, daily lifestyle, and social recreation. | ②③ | Cochrane Collaboration RoB | Exercise training has a positive effect on cognitive function in patients with mild cognitive impairment. |
| *Lin, Chen & Cheng (2023)* | 14/560 | RCT | Exercise interventions are not limited to exercise type, intensity, or duration | Participants in the control group either continued with their regular physical activities or engaged in sham exercises, such as stretching and balance activities. | ②③ | Cochrane Collaboration RoB | Walking has no significant benefit on cognitive function in individuals with mild cognitive impairment. |
| *Cai et al. (2023)* | 27/2,526 | RCT | physical and mental exercises such as taijiquan, Ba Duan Jin, qigong, meditation, yoga, music and dance | conventional care, health education or blank | ②③ | Cochrane Collaboration RoB | Physical and mental exercises has a positive effect on cognitive function in patients with mild cognitive impairment. |

**Notes.**

RCT (randomized controlled trial), NRCT (non-randomized controlled trial); ① for ADAS-cog, ② for MMSE, ③ for MOCA, ④ for COR-SOG,⑤ for NCSE.
**Table 2   The quality of AMSTAR2 items for included literatures.**

| | Q1 | Q2* | Q3 | Q4* | Q5 | Q6 | Q7* | Q8 | Q9* | Q10 | Q11* | Q12 | Q13* | Q14 | Q15* | Q16 |
|---|---|---|---|---|---|---|---|---|---|---|---|---|---|---|---|---|
| Ahn & Kim (2023) | Y | Y | N | Y | Y | Y | PY | Y | Y | N | Y | Y | Y | Y | Y | Y |
| Biazus-Sehn et al. (2020) | Y | Y | N | Y | Y | Y | PY | Y | Y | N | Y | N | N | Y | Y | Y |
| Gómez-Soria et al. (2022) | Y | Y | N | Y | Y | Y | PY | Y | Y | N | Y | N | N | Y | Y | N |
| Law et al. (2020) | Y | N | N | Y | Y | Y | PY | Y | Y | N | Y | N | N | Y | Y | Y |
| Pisani et al. (2021) | Y | N | Y | Y | Y | Y | PY | Y | Y | N | Y | N | N | Y | Y | Y |
| Song et al. (2018) | Y | N | N | Y | Y | Y | PY | Y | Y | N | Y | N | N | Y | PY | N |
| Song et al. (2022) | Y | N | N | Y | Y | Y | PY | Y | Y | Y | Y | N | N | Y | Y | N |
| Yu et al. (2021) | Y | N | N | Y | Y | Y | PY | Y | Y | Y | Y | N | N | Y | N | Y |
| Yuan, Li & Liu (2022) | Y | N | N | Y | Y | Y | PY | Y | Y | N | Y | N | N | Y | Y | Y |
| Zhang et al. (2019) | Y | N | N | Y | Y | Y | PY | Y | Y | N | Y | N | N | Y | Y | Y |
| Zhang et al. (2020) | Y | N | N | Y | Y | Y | PY | Y | Y | Y | Y | N | N | Y | N | Y |
| Zhou et al. (2020) | Y | Y | N | Y | Y | Y | PY | Y | Y | Y | Y | N | N | Y | Y | Y |
| Zhou & Li (2022) | Y | N | N | Y | Y | Y | PY | Y | Y | N | Y | N | N | Y | Y | Y |
| Zhu et al. (2020) | Y | N | N | Y | Y | Y | PY | Y | Y | Y | Y | N | N | Y | Y | Y |
| Zou et al. (2019) | Y | N | N | Y | Y | Y | PY | Y | Y | N | Y | N | N | Y | Y | Y |
| Han et al. (2023) | Y | Y | Y | Y | Y | Y | PY | Y | Y | Y | Y | N | N | Y | Y | Y |
| Lin, Chen & Cheng (2023) | Y | Y | Y | Y | Y | Y | PY | Y | Y | Y | Y | N | N | Y | Y | Y |
| Cai et al. (2023) | Y | Y | Y | Y | Y | Y | PY | Y | Y | Y | Y | N | N | Y | Y | Y |

Notes.

*Key entry

Y, Yes; PY, Partially yes; N, No; 1, Whether the research questions and inclusion criteria are detailed; 2, Provide preliminary design scheme (registration number); 3, Explain the reasons for the inclusion of the study type; 4, Use a comprehensive search strategy; 5, Two-person repeated literature screening was used; 6, Two-person repetitive data extraction was adopted; 7, Provide a list of exclusions and reasons; 8, Describe in detail the basic features included in the study; 9, Use appropriate tools to assess bias risk; 10, Report on funding sources included in the study; 12, Assess the impact of the risk of bias on the outcome; 13, Consider the risk of bias included in the study; 14, Heterogeneity of the findings was assessed; 15, Evaluate the possibility of publication bias; 16, Report relevant conflicts of interest.
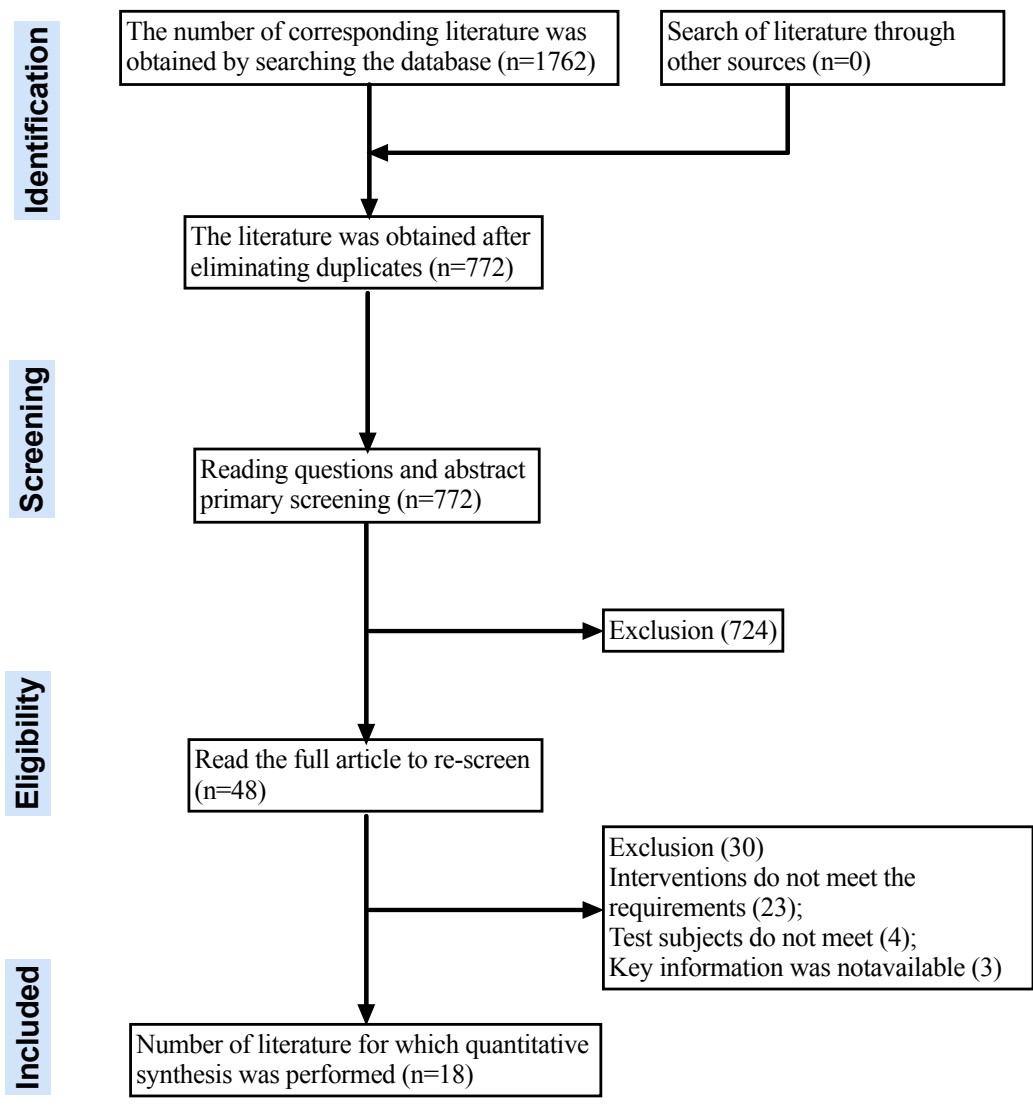

**Figure 1  PRISMA flow diagram chart for systematic review.**

exercise interventions on MMSE scores in patients with mild cognitive impairment, 13 of which showed that exercise was effective in improving MMSE scores, with only two not significantly different from controls.

### Alzheimer's disease assessment scale-cognitive subscale (ADAS-cog)

The ADAS-cog evaluates cognitive function in patients with MCI and MD and is second only to the MMSE in clinical trials. Seven studies (*Biazus-Sehn et al., 2020*; *Ahn & Kim, 2023*; *Zhou et al., 2020*; *Zhu et al., 2020*; *Pisani et al., 2021*; *Song et al., 2018*; *Zhang et al., 2020*) examined the effects of exercise interventions on ADAS-Cog scores in patients with mild cognitive impairment. All seven studies demonstrated that exercise effectively reduced ADAS-Cog scores.

### MoCA

The MoCA scale is a rapid screening tool for mild cognitive impairment (MCI) that assesses eight cognitive domains: attention and concentration, executive function, memory, language, visual-constructive skills, abstract thinking, calculation, and orientation. Eleven studies (*Biazus-Sehn et al., 2020*; *Song et al., 2022*; *Han et al., 2023*; *Lin, Chen & Cheng, 2023*; *Yuan, Li & Liu, 2022*; *Zhou et al., 2020*; *Zou et al., 2019*; *Cai et al., 2023*; *Song et al., 2018*; *Zhang et al., 2020*; *Yu et al., 2021*) investigated the effect of exercise interventions on MoCA scores in patients with MCI. Ten of these studies showed that exercise effectively improved MoCA scores, while one study found no significant effect.

### Neurobehavioral cognitive status examination (NCSE)

The NCSE is a standardized cognitive assessment scale that allows the initial screening and evaluation of patients based on their cognitive status. The NCSE covers the domains of orientation, concentration, language (comprehension, repetition, and naming), structural organization, memory, numeracy, and reasoning (similarity and judgment) and can more sensitively reflect problems in cognitive function and the degree of cognitive impairment. *Law et al. (2020)*, in examining the effects of an exercise intervention on NCSE scores in patients with mild cognitive impairment, showed that exercise was effective in improving NCSE scores.

### Chinese older adults' cognitive screening test (COR-SOG)

The COR-SOG scale was used to measure cognitive reserves. Cognitive reserve refers to an individual's ability to cope with cognitive decline and neurological impairment, which allows them to maintain better cognitive function in the face of cognitive impairment. *Law et al. (2020)*, in examining the effects of an exercise intervention on COR-SOG scores in patients with mild cognitive impairment, showed that exercise was effective in improving COR-SOG scores.

## DISCUSSION

Although several systematic reviews have confirmed the effects of exercise on cognitive function in older adults with MCI, few studies have explored the methodological quality of the corresponding studies or the certainty of evidence. All 18 meta-analyses were published after 2018, highlighting the recent and growing interest in this field. Notably, despite each meta-analysis employing corresponding risk-of-bias tools to assess the quality of the included original studies, the overall quality of evidence from these studies typically ranged from "very low" to "low." The limited quality of evidence considerably affects the reliability of current clinical recommendations regarding exercise interventions for mild cognitive impairment (MCI). This scenario underscores the need for cautious interpretation of the existing data and suggests that any recommendations made based on these meta-analyses should be regarded as preliminary.

### Methodological quality assessment of the meta-analytic literature

Methodological quality assessment of the 18 meta-analyses using the AMSTAR2 scale revealed that the overall methodological quality of the included trials was low. This

indicated that individual meta-analytic studies had varying degrees of methodological deficiencies in terms of topic selection, design, registration, data extraction, and statistical analysis. These deficiencies are mainly attributed to the following aspects.

(1) The included studies did not explain whether they had drafted a research proposal, and only a few studies were registered on the systematic review platform in advance. To effectively reduce the risk of selective reporting bias and avoid being affected by arbitrary decisions in data extraction and use, it is necessary to write a plan or pre-register in advance. In addition, registration on registration platforms (such as The Cochrane Library and PROSPERO) is also conducive to improving the transparency of meta-analysis research.

(2) None of the included studies provided a detailed description of the reasons for excluding the literature. Although the included studies reported a two-person search, screening, and data extraction, it is unknown whether this was performed with strict compliance, which increases the risk of bias. It is recommended that journals require authors to provide a list of excluded studies to ensure transparency during the screening process.

(3) The meta-analysis implementation process was not sufficiently rigorous. Seventeen studies did not adequately consider the potential impact of the risk of bias in the included studies on the analysis of results and did not provide satisfactory answers to the heterogeneity in the meta-analysis results. Additionally, three studies did not adequately evaluate publication bias. Meta-analysis is recognized as the best method by which research evidence on a particular issue can be objectively evaluated and synthesized. The risk of bias analysis, interpretation of sources of heterogeneity, and evaluation of publication bias during the implementation of the meta-analysis determined the reliability of the results. When randomized controlled trials or non-randomized intervention studies with different risks of bias are included, the investigator should conduct an in-depth analysis of the risk of bias in the included literature and discuss its impact on the results (qualitative analysis results or pooled effect sizes). In addition, when heterogeneity tests suggest the presence of heterogeneity, researchers should make every effort to find potential causes of heterogeneity and explore the probable causes of heterogeneity in the discussion section.

(4) Ten studies did not specify the funding sources for the studies included in the meta-analysis production process, which may have led to potential bias, opacity, and weakened trust. The source of funding for a study may have an impact on the findings, as funders may have their own interests and preferences. Lack of information on the source of funding may cause readers to doubt the reliability and transparency of the study. Knowing the source of funding can help readers to understand the context of the study and the interests of the researcher. Simultaneously, if readers suspect that the results of the study are influenced by the source of funding, they may be skeptical about the results of the study and reduce trust in the study.

These findings have significant implications for future research methodologies and clinical practices in exercise interventions for MCI. Our results underscore the importance of adhering to rigorous research protocols and pre-registering studies to enhance transparency and replicability in the design and implementation of exercise interventions

for MCI. Detailed documentation of excluded literature and clear disclosure of funding sources will contribute to the credibility of the research and public trust in the findings. More stringent analyses of bias risk and heterogeneity in the studies included in the meta-analysis will improve the accuracy and practicality of exercise intervention recommendations in clinical practice.

## Interpretation of heterogeneity in included studies

In conducting the methodological quality assessment of the included meta-analyses, we observed significant heterogeneity among the studies, which notably impacted our evaluation results. The heterogeneity manifested in several key aspects.

(1) Diversity in study design

The meta-analyses varied in their underlying study designs, encompassing differences in the types of exercise interventions, baseline cognitive status of participants, and the duration and frequency of the interventions. This diversity could lead to varying interpretations of the effects of exercise interventions.

(2) Participant characteristics

Variations in participant demographics such as age, gender, and stage of disease could influence the effectiveness of the exercise interventions. Different studies may focus on distinct populations, affecting the perceived efficacy of exercise on cognitive functions.

(3) Methodological variability

Inconsistencies in data extraction, statistical analysis approaches, and bias risk assessment across studies could contribute to variability in conclusions. For instance, the methods for handling missing data and the choice of effect size models varied, potentially leading to different outcomes.

In summary, the heterogeneity of the studies played a significant role in our methodological assessment. Future methodological evaluations of similar meta-analyses should delve deeper into these heterogeneity factors, detailing them in the study design and reporting to enhance the clarity and comprehensiveness of the findings.

## Recommendations for improvement

Based on the results of this study, the author makes the following recommendations were made to inform future research and evaluation criteria:

(1) Meta-analysis authors should adhere to the appropriate guidelines for study design, implementation, and reporting. Essential strategies include (a) Consulting the Cochrane Handbook for Systematic Reviews of Interventions to grasp meta-analysis fundamentals, applications, and advantages/disadvantages of relevant software (*e.g.*, RevMan, Stata, R, CMA, and OpenMetaAnalyst). For instance, despite RevMan's suitability for exercise intervention trials, it has limitations in publication bias testing, and authors should be familiar with comprehensive tests such as Egger's regression analysis and Begg's rank correlation method available in CMA and Stata. Additionally, increasing the pre-registration of protocols is vital. Authors are encouraged to pre-register their study protocols in registries such as PROSPERO, detailing the planned methods and analyses to enhance transparency and aid in reducing publication bias. Furthermore, enhancing the

thorough reporting of study screening and reasons for exclusion is critical for improving transparency and reproducibility. Authors must provide detailed accounts of study selection processes, including clear documentation of screening procedures, criteria for inclusion and exclusion, and explicit reasons for excluding studies, especially those that might influence the meta-analysis outcomes. Better handling of heterogeneity is also essential. Authors should not only report the presence of heterogeneity but explore and explain its sources. This can be achieved by conducting subgroup analyses or meta-regression to investigate potential moderators of effect sizes. A detailed interpretation of how variations between studies might impact the overall findings and conclusions will significantly enhance the quality and credibility of the meta-analysis. (b) Refining and pre-registering study plans based on AMSTAR2 and PRISMA checklists, ensuring detailed reporting, particularly in the AMSTAR2 Methods section, to minimize bias. (c) Emphasizing both the theoretical and practical implications of meta-analyses, high-quality studies should provide reliable evidence, adhere to reporting standards, and advance research fields while offering practical guidance.

(2) To improve the quality of meta-analyses, journal editorial boards and departments should strengthen the reporting requirements through review policies. We suggest the following measures: (a) Develop self-checking report guidelines for meta-analysis authors based on AMSTAR2 and PRISMA checklists tailored to specific situations and establish meta-analysis topics to enhance their impact and standardization. (b) Journal editors should refer to meta-analysis reporting guidelines when initially screening manuscripts or conducting peer-expert reviews to ensure the accuracy and coherence of the report. (c) It should be noted that the AMSTAR2 and PRISMA checklists mainly apply to trial (intervention)-type meta-analyses; some entries may not be applicable to meta-analyses of sports science correlation studies.

(3) Universities need to enhance their theoretical and practical courses on systematic reviews and meta-analyses. While some medical schools or comprehensive colleges have already introduced ''evidence-based medicine'' or ''systematic review/meta-analysis'' courses, sports colleges often lack such offerings, hindering the development of professionals in the field. Therefore, we suggest that sports colleges and universities incorporate meta-analysis theory and practice courses into their postgraduate training programs or integrate relevant lectures on systematic review/meta-analysis topics into existing ''research methods'' and ''statistics'' courses. This approach can better cultivate professionals with systematic review/meta-analysis expertise and improve the overall educational quality.

## Limitations

There are some limitations to this study: (1) only English literature was included, and the search results may not be comprehensive; (2) there is inevitably some subjectivity in the evaluation and analysis process, which may also lead to bias; (3) a lack of discrimination capacity of the AMSTAR 2.0 tool may affect the ability to achieve moderate or high methodological quality in systematic reviews; (4) the NCSE and COR-SOG scales were only administered once, and the results should be treated with caution.

## CONCLUSION

Meta-analyses of exercise interventions for mild cognitive impairment generally exhibit quality levels ranging from "very low" to "low." Key deficiencies are noted in the registration of preliminary design plans, transparency in the rationale for including specific study types, and the assessment of bias risks that significantly affect the outcomes—most studies fail to meet the requisite standards in these areas. Additionally, the widespread lack of reporting on funding sources could lead to undeclared biases and conflicts of interest, potentially compromising the interpretation and credibility of the findings. Overall, the existing research lacks methodological rigor, and the quality of reporting is notably poor.

**Abbreviations**

| | |
|---|---|
| **MCI** | Mild Cognitive Impairment |
| **AD** | Alzheimer's Disease |
| **GABA** | Gamma-Aminobutyric Acid |
| **WMS** | Wechsler Memory Scale |
| **MMSE** | Mini-Mental State Examination |
| **MoCA** | Montreal Cognitive Assessment |
| **ADAS-cog** | Alzheimer's Disease Assessment Scale-Cognitive Subscale |
| **NCSE** | Neurobehavioral Cognitive Status Examination |
| **COR-SOG** | Chinese Older Adults' Cognitive Screening Test |
| **BDNF** | Brain-Derived Neurotrophic Factor |
| **VEGF** | Vascular Endothelial Growth Factor |

### Funding

This work was funded by the Basic Scientific Research Business Fee Project of Provincial Undergraduate Universities in Heilongjiang Province (2021KYYWF-FC02). The funders had no role in study design, data collection and analysis, decision to publish, or preparation of the manuscript.

### Grant Disclosures

The following grant information was disclosed by the authors:
Basic Scientific Research Business Fee Project of Provincial Undergraduate Universities in Heilongjiang Province: 2021KYYWF-FC02.

### Competing Interests

The authors declare there are no competing interests.

### Author Contributions

- Wanli Zang conceived and designed the experiments, performed the experiments, analyzed the data, prepared figures and/or tables, authored or reviewed drafts of the article, and approved the final draft.

- Qinghai Zou conceived and designed the experiments, analyzed the data, prepared figures and/or tables, and approved the final draft.
- Ningkun Xiao performed the experiments, analyzed the data, prepared figures and/or tables, and approved the final draft.
- Mingqing Fang performed the experiments, analyzed the data, prepared figures and/or tables, and approved the final draft.
- Su Wang conceived and designed the experiments, performed the experiments, prepared figures and/or tables, authored or reviewed drafts of the article, and approved the final draft.
- Jingjing Chen conceived and designed the experiments, performed the experiments, authored or reviewed drafts of the article, and approved the final draft.

## Data Availability

This is a systematic review/meta-analysis.

## Supplemental Information

Supplemental information for this article can be found online at http://dx.doi.org/10.7717/peerj.17773#supplemental-information.

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
