# Peer review of "A methodological and reporting quality assessment of systematic reviews/meta-analyses on exercise interventions for cognitive function in older adults with mild cognitive impairment"

_PeerJ, doi:10.7717/peerj.17773_

## Round 0.1 · original submission · Major Revisions

I congratulate the authors on their well written manuscript. Before publication, several points need to be addressed which are raised by the reviewers. This includes the AMSTAR methodology as well as the general methodological approach (e.g., search strategy)

·

Basic reporting

The introduction effectively provides context on the epidemiology of MCI, rationale for considering exercise as a non-pharmacological intervention, and the study’s objectives. The manuscript structure adheres well to PeerJ standards. Tables and Figure are relevant and clearly described. Two minor issues should be considered:
1. One discussion sub-section (4.4 Evidence-based exercise prescription formulation; lines 316-378) seems not related to the purpose of this review. Please clarify its relevance to the study’s objectives; otherwise, I recommend removing this section. Also, this section was not well referenced.
2. Please ensure thorough proofreading of the manuscript. For example, on line 31, it was written as “the effects of exercise on cognitive function interventions in …”, where it seems “interventions” should follow “exercise”. On Line 100, it should be “six” databases instead of “five”; the “Physiotherapy evidence database” was not included in the Abstract.

Experimental design

The study was pre-registered, and its sections on search strategy, eligibility criteria, literature selection, data extraction, and methodological quality evaluation sections adhere well to the PRISMA guidelines.

Major issues:
1. Search strategy (line 102): The database search was performed on Feb 23, 2023, which was a year ago. I recommend updating the search to ensure the inclusion of the most recent literature.
2. GRADE assessment: The authors used GRADE to assess the certainty of evidence for each cognitive outcome in each meta-analysis and summarized the evidence using proportions. However, it's recommended to integrate and assess evidence from meta-analyses targeting the same cognitive domain. Please refer to item 14 of the PRIOR checklist (Gates et al., 2022) and the GRADE guidelines (Guyatt et al., 2013). Explicitly outlining the criteria to assess each domain is recommended, as shown in Box 2 of Law et al. (2020). However, applying GRADE in this context should be approached cautiously, as its adaptation to overviews of systematic reviews lacks clear guidance. As mentioned in Holy (2021), "The GRADE approach, although very useful and widely accepted as a means of rating the body of evidence originating from single primary studies or systematic reviews, is not directly transferable to overviews of systematic review" (p.70). Bougioukas et al. (2019), "there is shortness of guidance on how to apply GRADE within an overview and this approach should be followed with caution". I won't recommend applying GRADE in this quality assessment of meta-analyses unless you found well-established guidance on the adaptation of GRADE to overviews.

Minor issues:
1. Previous quality assessments with similar interventions, outcomes, populations, and using the same assessment tool (i.e., AMSTAR2) should be acknowledged and described, clarifying how the current quality assessment compares with or advances earlier ones (e.g., Quan et al., 2024 and Venegas-Sanabria et al., 2021).
2. Inclusion criteria (Lines 119-120): “The included STUDIES were required to have at least one intervention group engaged in exercise interventions.” However, since this is an assessment of meta-analyses, the inclusion criterion for interventions should relate to meta-analyses rather than individual studies.
3. Data extraction (Line 142-144): it appears that primary outcome measures are all about global cognition. However, the inclusion criteria didn't specify that the outcome should be limited to global cognition. To my knowledge, some included meta-analyses reported effect sizes for other cognitive domains, such as memory and executive function (e.g., Biazus-Sehn, 2020). Therefore, please specify your outcome of interest (i.e., global cognition) in the inclusion criteria, objectives, and title.

References
Bougioukas, K. I., Bouras, E., Apostolidou-Kiouti, F., Kokkali, S., Arvanitidou, M., & Haidich, A. B. (2019). Reporting guidelines on how to write a complete and transparent abstract for overviews of systematic reviews of health care interventions. Journal of Clinical Epidemiology, 106, 70-79.
Gates, M., Gates, A., Pieper, D., Fernandes, R. M., Tricco, A. C., Moher, D., ... & Hartling, L. (2022). Reporting guideline for overviews of reviews of healthcare interventions: development of the PRIOR statement. Bmj, 378.
Guyatt, G., Oxman, A. D., Sultan, S., Brozek, J., Glasziou, P., Alonso-Coello, P., ... & Schünemann, H. J. (2013). GRADE guidelines: 11. Making an overall rating of confidence in effect estimates for a single outcome and for all outcomes. Journal of clinical epidemiology, 66(2), 151-157.
Holly, C. (2021). Umbrella reviews. Comprehensive systematic review for advanced practice nursing, 373-381.
Law, C. K., Lam, F. M., Chung, R. C., & Pang, M. Y. (2020). Physical exercise attenuates cognitive decline and reduces behavioural problems in people with mild cognitive impairment and dementia: a systematic review. Journal of physiotherapy, 66(1), 9-18.
Quan, Y., Lo, C. Y., Olsen, K. N., & Thompson, W. F. (2024). The effectiveness of aerobic exercise and dance interventions on cognitive function in adults with mild cognitive impairment: an overview of meta-analyses. International Review of Sport and Exercise Psychology, 1-22.
Venegas-Sanabria, L. C., Martínez-Vizcaino, V., Cavero-Redondo, I., Chavarro-Carvajal, D. A., Cano-Gutierrez, C. A., & Álvarez-Bueno, C. (2021). Effect of physical activity on cognitive domains in dementia and mild cognitive impairment: overview of systematic reviews and meta-analyses. Aging & Mental Health, 25(11), 1977-1985.

Validity of the findings

The Results and Discussion sections summarized the methodological quality assessment of the included meta-analyses and provided recommendations for future systematic reviews based on the findings. It’s also a great practice to interpret the heterogeneity of the included studies.

Major issue:
The quality was assessed using the AMSTAR2. Please carefully read the detailed explanation for each item in the development paper of AMSTAR2 (Shea BJ et al., 2017) and its supplementary "AMSTAR2 guidance document". The online checklist provided by AMSTAR is also helpful (https://amstar.ca/Amstar_Checklist.php). I found the following items should be carefully checked.
1) item 7: This item requires review authors to provide a complete list of potentially relevant studies with justification for the exclusion of each one. To my knowledge, most of the studies didn't include such a list. For example, Ahn (2023) excluded 64 papers during full-text screening but did not provide an appendix that lists the 64 excluded papers and reasons for exclusion.
2) item 10: The review authors should document the funding sources for each study included in the review. Also taking Ahn (2023) as an example, they did not detail the funding sources for each included study in Table 1.
3) item 12 and 13: For item 12, when including RCTs of variable quality, the review should assess the impact by regression analysis or by estimating pooled effect sizes with only studies at low ROB. For item 13, review authors should include a discussion of the impact of ROB in the interpretation of the results of the review. However, for example, Zhang et al. (2020) may not fulfill these two items. Even if ignoring the blinding item (given that it's impractical to blind participants and personnel in an exercise intervention), one of their included studies - Lu (2015) – was found to have a high risk of "selective reporting". To meet the criterion for item 12 in AMSTAR2, Zhang et al. (2020) should at least perform a sensitivity analysis excluding Lu (2015) to assess the impact of risk of bias on the overall effect sizes.

Minor issues:
1. Results (Lines 165-166): Recommend including a list of excluded studies and reasons for exclusion during full-text screening, as recommended by the AMSTAR2.
2. The conclusion is not closely linked to its objective of assessing the methodological quality of meta-analyses. It only has one beginning sentence to summarize the findings, the other parts mainly discuss recommendations for future meta-analyses. Authors should specify aspects that lead to the conclusion that “studies are not rigorous enough.”
3. According to Table 1, the included study (Wei, 2020) included quasi-experimental studies, not exclusively RCTs, which may not meet the inclusion criterion.
4. Results (line 193): Of the 14 studies assessing MMSE, based on the description of the results, “12 of which showed that exercise was effective in improving MMSE scores, with only one not significantly different from controls”. I suppose there should be another study not being summarized.

Reference
Shea, B. J., Reeves, B. C., Wells, G., Thuku, M., Hamel, C., Moran, J., ... & Henry, D. A. (2017). AMSTAR 2: a critical appraisal tool for systematic reviews that include randomised or non-randomised studies of healthcare interventions, or both. bmj, 358.

Reviewer 2 ·

Basic reporting

See below

Experimental design

See below

Validity of the findings

See below

Additional comments

The authors have conducted a metahodological assessment and quality appraisal of systematic reviews and meta-analyses examining the effects of exercise interventions on cognitive function in older adults with mild cognitive impairment (MCI). This is an important topic, as exercise shows promise as a non-pharmacological approach for managing MCI. Evaluating the quality of existing meta-analyses can help identify areas for improvement in research on this topic.
The methodology is clearly described, and the authors used appropriate tools, including AMSTAR2 for assessing methodological quality and GRADE for evaluating certainty of evidence. The results are well-presented, showing that most of the included meta-analyses were of low or very low methodological quality, with the certainty of evidence also being low for most outcomes.
A few suggestions for improvement:
1. In the Discussion, expand on the implications of the poor quality/certainty of evidence. How does this impact the ability to make clinical recommendations regarding exercise for MCI based on existing meta-analyses?
2. Consider providing more specific guidance on how future meta-analyses could be improved, such as increased pre-registration of protocols, more thorough reporting of study screening and reasons for exclusion, better handling of heterogeneity, etc.
3. The English language is generally good but there are some grammatical issues and awkward phrasing throughout. I recommend careful proofreading and editing to improve clarity and readability.

Overall, this is a well-conducted methodological assessment on an important topic. With some expansion of the Discussion and improvements in writing, it can make a valuable contribution to guiding future research on exercise interventions for MCI.

---

## Round 0.2 · Minor Revisions

I thank the authors for providing feedback to the reviewers comments. While reviewer #2 is happy to accept the paper, reviewer #1 does have some more valid comments which need to be addressed.

·

Basic reporting

Minor issues:
1. Please cross-check the references for accuracy (e.g., reference 20 is listed as the GRADE guideline, but it refers to AMSTAR2 in the text on line 86).
2. In Section 3.4, it appears that you are summarizing the results for each cognitive measure but not evaluating the certainty of evidence. Please consider changing the heading to something more accurate. Additionally, as the certainty of evidence has not been assessed, please explain why Table 2 "Certainty of Evidence of Included Literatures" was still included.

Experimental design

Major issue: I cannot see you have comprehensively searched the databases and assessed their eligibility. I searched EMBASE using your search string for papers published between 2023 and the present. Upon roughly reviewing the results, I found two papers that meet your inclusion criteria: (1) "Effects of mind-body exercise on cognitive performance in middle-aged and older adults with mild cognitive impairment: A meta-analysis study," and (2) "Efficacy of non-pharmacological intervention on cognitive function of elderly patients with mild cognitive impairment: a network meta-analysis." However, these were not included in your reference list or the appendix of excluded papers.

Minor issue: Regarding my previous comment on studies that assessed quality using AMSTAR2 with similar PICOS, I meant that some umbrella reviews (e.g., Quan et al., 2024, and Venegas-Sanabria et al., 2021) also used the same quality assessment tool (i.e., AMSTAR2) to assess previous meta-analyses on similar populations (older adults with mild cognitive impairment), interventions (exercise), and outcomes (cognition). Please clarify your differences and innovations compared to these previous umbrella reviews. According to PRISMA 2020, "If other systematic reviews addressing the same (or a largely similar) question are available, explain why the current review was considered necessary. If the review is an update or replication of a particular systematic review, indicate this and cite the previous review."

Validity of the findings

Major issue: While you have revised the AMSTAR2 assessment results in Table 3, the corresponding discussion section (i.e., Section 4.1) has not been updated regarding the assessment outcomes. For example, in the third point, there are 16 studies instead of two studies "did not adequately consider the potential impact of the risk of bias." In the fourth point, there are actually 10 studies that "did not specify the funding sources." These are just examples from a rough scan; please carefully check for consistency between the results in Table 3 and the summary of the results.

Minor issues: The Appendix: Even though it is an appendix, I still recommend citing the excluded papers in a proper referencing format (at least to have the first author’s name, year of publication, and journal). Additionally, the reasons for exclusion have not been included. A table with the first column for the reference and the second column for the reasons for exclusion might be helpful.

Reviewer 2 ·

Basic reporting

None

Experimental design

None

Validity of the findings

None

Additional comments

All my concerns have been addressed.

---

## Round 0.3 · accepted · Accept

The authors have addressed all of the reviewers' comments. Congratulations. The manuscript is ready for publication

·

Basic reporting

All my concerns have been addressed.

Experimental design

All my concerns have been addressed.

Validity of the findings

All my concerns have been addressed.